# Predicting the frequency of positive laboratory submissions for porcine reproductive and respiratory syndrome in Ontario, Canada, using autoregressive integrated moving average, exponential smoothing, random forest, and recurrent neural network

Tatiana Petukhova[1]*, Maria Spinato[2], Tanya Rossi[2], Michele T. Guerin[1],
Cathy A. Bauman[1], Pauline Nelson-Smikle[2], Davor Ojkic[2], Zvonimir Poljak[1]

**1** Department of Population Medicine, Ontario Veterinary College, University of Guelph, Guelph, Ontario, Canada, **2** Animal Health Laboratory, Laboratory Services Division, University of Guelph, Guelph, Ontario, Canada

☻ These authors contributed equally to this work
* tpetukho@uoguelph.ca

## Abstract

Porcine Reproductive and Respiratory Syndrome Virus (PRRSV) is endemic in many pig-producing countries and poses significant health and economic challenges. Enhanced surveillance strategies are essential for effective disease management. This study aimed to evaluate and compare the performance of different time-series modeling techniques to predict weekly PRRSV-positive laboratory submissions in Ontario, Canada. Ten years of PRRSV diagnostic data were obtained from the Animal Health Laboratory at the University of Guelph and were processed into a weekly time series. The dataset was analyzed with autoregressive integrated moving average (ARIMA), exponential smoothing (ETS), random forest (RF), and recurrent neural network (RNN) models. Two validation strategies were employed: a traditional train-test split and a simulated prospective rolling forecast. Model accuracy was evaluated using common predictive error metrics. Descriptive analysis indicated a gradual increase in PRRSV positive submissions over time, with no consistent seasonal pattern. ARIMA and ETS models generally overpredict case counts, while RF and RNN tended to underpredict them. Among the evaluated models, the RF regression model most accurately captured the underlying time-series dynamics and produced the lowest prediction errors across both validation approaches. Despite outperforming other models, the RF model's high relative prediction errors limit its suitability for accurate forecasting of PRRSV-positive submissions in Ontario's routine surveillance system. Further data refinement and algorithm improvements are warranted.

**Data availability statement:** Data used for this study cannot be shared publicly due to our REB application. Part of the REB application is also founded on the confidentiality agreement between the principal investigator and the laboratory providing access to the data. They are confidential and, in some cases, could be identifiable even after de-identification. Contact information for our institutional REB committee is reb@uoguelph.ca.b.

**Funding:** The author who received funding: ZP Grant number: 499163 This project was funded by the University of Guelph's Food From Thought program The author who received funding: MS Grant number: 499170 This project was funded by the University of Guelph's Food From Thought program The author who received funding: ZP Grant number: 10002 This project was funded by the Ontario Agri-Food Innovation Alliance The funders had no role in study design, data collection and analysis, decision to publish, or preparation of the manuscript.

**Competing interests:** The authors have declared that no competing interests exist.

## Introduction

Porcine Reproductive and Respiratory Syndrome Virus (PRRSV) is an endemic pathogen with almost global distribution. The virus is classified into two types: Type I, initially detected and still the dominant type in Europe [1–4], and type II, originally detected and most frequently detected in North America [5,6]. The high herd-level incidence and prevalence in commercial swine operations in North America are driven by multiple factors. These include the epidemiological characteristics of PRRSV infection, and the inherently dynamic demographic characteristics of swine populations. PRRSV outbreaks continue to occur despite implementation of good management practices and external and internal biosecurity measures aimed at preventing infection or successfully eliminating the virus from affected populations. Due to the nature of PRRSV infection and multiple transmission pathways, both at the individual and at the herd-level, decision-making is dependent on regular assessment of PRRSV infection status. Thus, many herds are tested regularly for the presence of PRRSV circulation, and this represents a rich data source. Routine testing conducted in diagnostic laboratories constitutes a key component of a robust monitoring system. Data generated from such testing can serve multiple purposes, including descriptive trend analysis, identification of periods in which observed values significantly deviate from expected behavior, and forecasting predictive applications.

Descriptive dashboards have been created for end-users where the results of PRRSV testing are summarized, displaying frequency of submissions, positive submissions, molecular typing, and other statistics of interest [7]. Accurately predicting the frequency of positive submissions using time-series models could provide a valuable complement to the entire surveillance ecosystem for PRRSV. However, before incorporating such a forecasting tool, it is essential to compare relevant state-of-the-art forecasting algorithms and assess if the observed accuracy is satisfactory for field deployment.

Previous studies, including those by Poljak et al. [8] and Petukhova et al. [9], have suggested that diagnostic test results from herd submissions may exhibit temporal structures such as trends, seasonality, or cyclic patterns. These data are appropriately analyzed using time-series models, which facilitate, identify and interpret underlying temporal patterns, forecast future outcomes, and assess historical behavior.

The objective of this study is to compare the forecasting performance of two classical statistical methods – autoregressive integrated moving average (ARIMA) and exponential smoothing (ETS) – with one machine learning technique, random forest (RF), and one deep learning approach, recurrent neural network (RNN). The goal is to identify the most accurate method for potential integration into a routine surveillance system, such as interactive animal pathogen dashboards (IAPD).

## Materials and methods

### Data

Data on PRRSV-positive submissions were obtained from the Animal Health Laboratory (AHL) at the University of Guelph for the period from January 2011 to July

2023, based on submissions from swine farms in Ontario, Canada. Each submission contained one or more samples, and all samples were tested with PRRSV RT-PCR. Overall, the AHL received 152,723 samples in 31,349 unique swine submissions. Of those submissions, 2,122 samples in 2,055 unique submissions were also evaluated using sequence analysis of the open reading frame-5 (ORF-5) and were, therefore, excluded from data processing to avoid duplication. The remaining 150,601 samples in 31,318 unique submissions were tested for different purposes: 120,598 samples in 26,164 unique submissions (83.5%) were tested for diagnosis; 28,090 samples in 5,055 unique submissions were tested for monitoring (16.1%); 1,899 samples in 96 unique submissions (0.3%) were tested for research; 5 samples in 2 unique submissions (0.006%) were tested for financial purpose; and 9 samples in one submission (0.003%) were tested for other (not specified) purpose. Submissions tested for research and financial purposes were excluded based on the rationale that they might not represent actual clinical PRRS disease in a herd. The remaining 148,697 samples in 31,220 unique submissions underwent further selection based on reported test results. The reported results of tested samples were qualitative (e.g., positive, negative, inconclusive, suspicious, not detected, no result, not analyzed). The results interpreted as "no result" or "not analyzed" were treated as missing values, and corresponding samples were excluded from the computation procedure [7,10], resulting in 148,645 samples in 31,213 unique submissions. The format of displaying values was converted into a dichotomous outcome: test results reported as "negative", "inconclusive", "suspicious", or "not detected" were considered negative; otherwise, they were considered positive. Test results were aggregated at the submission level. A submission was treated as positive if at least one sample within the submission was declared positive. The records were cleaned, anonymized, and aggregated. Submission dates were converted into weekly intervals, with each week starting on Sunday. To maintain a consistent frequency across study years and to keep all records, the number of tested submissions and the number of positive submissions were adjusted for 53-week-years. The 53rd week was merged into the following year's first week (Week 1). Weekly counts of PRRSV-positive submissions were analyzed using time-series methods.

All data processing and analyses were conducted in R version 4.1.1 [11].

## Statistical methods

The time series of weekly PRRSV-positive submissions was initially explored graphically to gain an overview of its structure. The distribution of weekly counts in each week across 13 years (2011–2023) was assessed using boxplots generated with the `boxplot()` function from the *graphics* package. Stationarity – defined as the property of a constant mean and variance over time – was assessed both visually and formally. Visual inspection involved autocorrelation and partial autocorrelation plots, generated using the `acf()` and `pacf()` functions from the *stats* package. Formal testing was conducted using the Augmented Dickey-Fuller (ADF) unit root test, implemented via the `ur.df()` function from the *urca* package [12].

Time-series data were further analyzed in terms of three components: trend-cycle, seasonality, and remainder (residuals). To identify underlying temporal patterns and guide the choice of a forecasting approach, the time series was decomposed into these three components using the `stl()` function from the *stats* package [13]. This approach employs locally weighted regression (Loess) to capture nonlinear trends and seasonal structures in the data. The trend-cycle window parameter (`t.window`) in the `stl` function specifies the number of consecutive observations used to estimate the trend component, thereby controlling its smoothness. For this analysis, `t.window` was set to 21 weeks. The seasonal window parameter (`s.window`) determines the number of consecutive observations used to estimate the seasonal component and controls the degree of smoothing applied to seasonal fluctuations. As the seasonal pattern was assumed to be constant across years, `s.window` was set to "`periodic`" which replaces local smoothing with averaging across corresponding periods.

Following both graphical and formal assessments, the time series of weekly PRRSV-positive submissions was modelled using two statistical approaches (ARIMA and ETS), one machine learning method (RF), and one deep learning

technique (RNN). These models were selected to exploit their respective strengths in capturing linear, seasonal, and nonlinear patterns, with the goal of improving forecast accuracy.

Residual diagnostics were conducted where applicable to assess the model's adequacy. The final models were used to predict the frequency of PRRSV cases in the swine population of Ontario. Predictive performance was evaluated with two validation approaches that are described in detail in the validation methods subsection.

### Autoregressive integrated moving average (ARIMA)

First, the time series of weekly PRRSV-positive submissions was modeled using ARIMA models. ARIMA is a widely used approach for time-series forecasting, developed to capture the autocorrelation structure of data. The model-building process involves several key steps: model identification, parameter estimation, model selection, diagnostic checking (model validation), and forecasting.

Model identification involves determining the appropriate orders of the autoregressive (AR), integrated (I), and moving average (MA) components, along with estimating their associated parameters. The idea for this approach is to transform a non-stationary time series into a stationary one by addressing trend-cycle components and seasonal fluctuations. A stationary time series is prerequisite for reliable ARIMA modeling. Further details of the ARIMA modeling are available in the Supporting Information (S1 File).

The process of ARIMA model identification and fitting can be computationally intensive, particularly when the data exhibit intricate autocorrelation structures. To facilitate this process, the stepwise model selection algorithm implemented in the `auto.arima()` function from the *forecast* package [14] was employed, and the Box-transformation parameter was set to "auto". This allowed the function to automatically select an appropriate variance-stabilizing transformation based on the data, helping to improve model performance [15]. Model selection was guided according to minimization of the Akaike's Information Criterion (AIC), ensuring an optimal balance between goodness-of-fit and model complexity.

### Exponential smoothing (ETS)

Exponential smoothing (ETS) models, similar to ARIMA models, are extensively employed for time-series analysis and forecasting. While both approaches provide complementary information, they differ in their underlying assumptions and modeling strategies. ARIMA models are primarily used for analyzing stationary and non-stationary time series, with differencing applied to achieve stationarity when necessary. In contrast, ETS models are particularly well-suited for non-stationary time series and explicitly model trend and seasonal components. Forecasts are generated using exponentially weighted averages of past observations, with larger weights placed on recent data and exponentially decreasing over time. Previous studies have demonstrated the flexibility of ETS models in capturing complex seasonal and trend patterns [16–18].

The ETS (Error, Trend, Seasonal) class of models represents an exponential smoothing forecasting framework, also known as innovations state space models. Each ETS model is defined by the specification of its trend, seasonal, and error components. The seasonal and error components can be either none, additive or multiplicative, while the trend component can be specified as none, additive, or additive damped. The models are characterized by their forecasting equations and associated smoothing parameters. A detailed description of the ETS modeling framework is provided in the Supporting Information in (S1 File).

An ETS model was identified using the stepwise model automatic procedure implemented in the `ets()` function from the *forecast* package [14]. The modeling framework follow the notation described in [19,20], where a model is denoted by a three-character string representing the error type, trend type, and seasonal components, respectively. We did not predefine a specific combination of these components; instead, the model structure was selected automatically by setting the specification to (Z, Z, Z), allowing the procedure to determine the optimal configuration based on information criteria.

The Box-Cox transformation parameter was automatically estimated by setting the lambda argument to "auto" [15]. Model selection was based on the lowest Bayesian Information Criterion (BIC), which balances goodness-of-fit with parsimony.

## Random forest (RF)

Previous studies have demonstrated that the RF algorithm can effectively model the dynamics of count data [9]. Accordingly, we employed this approach to predict the number of PRRSV-positive submissions. It is important to note that the RF algorithm assumes observations are independent and identically distributed and does not inherently account for temporal dependencies present in time-series data. To address this limitation, we pre-processed the data by applying differencing and statistical transformations to stabilize the mean and variance over time. Furthermore, we employed time-delay embedding to restructure the temporal data into a supervised learning format, thereby enabling the model to capture linear or nonlinear relationships between past and present observations.

The required order of seasonal differencing was estimated using the `nsdiffs()` function from the *forecast* package [14]. To identify an appropriate power transformation for improving normality, Tukey's Ladder of Powers transformation was applied using the `transformTukey()` function from the *rcompanion* package [21]. However, no suitable transformation was identified, and therefore, the time series was retained on its original scale.

To capture temporal dependencies in the data, time-delay embedding was applied. This technique reconstructs a time series in a low-dimensional Euclidean space defined by the embedding dimension *K*. Specifically, the weekly PRRSV-positive submission time series was transformed into a matrix of overlapping lagged observations using the `embed()` function from the *stats* package [22].

Based on the results from prior ARIMA analysis, a lag of 12 weeks was selected for time-embedding. The embedding dimension was set to 13 (i.e., one greater than the number of lags), resulting in a matrix where the first column represented the target variable at time *t*, and the remaining 12 columns corresponded to the lagged values from *t-1* to *t-12*, serving as predictors.

Integrating time-delay embedding into the RF procedure transformed the time-series forecasting problem into a supervised regression task, where past observations served as input features to predict feature values.

RF regression was implemented using the `randomForest()` function from the *randomForest* package [23,24]. Hyperparameter optimization was performed with the tuneRF() function from the same package, which iteratively searches the number of variables randomly sampled at each split (mtry) and the number of trees (ntree) that minimize the out-of-bag (OOB) error rate. The optimal parameter values were subsequently integrated into the final RF regression model. Further details of the RF regression procedure are available in the Supporting Information (S1 File).

## Recurrent neural Networks (RNN)

To explore flexible statistical methods capable of capturing potential nonlinear regression effects, we employed RNN models, which are well-suited for modeling complex nonlinear relationships between a response variable and its predictors. RNNs, initially proposed by J. Hopfield in 1982 [25], constitute a class of deep learning models specifically designed to handle sequential or time-series data. They are trained to learn temporal dependencies and generate sequential forecasts. An RNN can be conceptualized as a network of interconnected neurons organized in layers, where recurrent connections allow information to be fed back into the network. The output at each time-step depends on both the current input and the previous hidden states of the sequence, enabling the model to capture temporal patterns.

Although standard RNNs are capable of modeling sequential data, they often struggle to retain information from earlier time-steps, as the contribution of earlier inputs can dimmish significantly during the learning process, a phenomenon known as the vanishing gradient problem [25]. To address this limitation, Hochreiter and Schmidhuber proposed the Long Short-term Memory (LSTM) architecture, which introduces memory cell and filters, known as gating mechanisms, that

regulate what information is retained, updated or discarded over time [26]. The Gated Recurrent Unit (GRU), a simplified variant introduced by Cho et al. [27], combines the functions of forgetting old information and incorporating new information into a single mechanism called the update gate, thereby reducing computational complexity while maintaining comparable performance. Further details about the RNN are available in the Supporting Information (S1 File).

Prior to applying RNN, the time series was reshaped into a matrix format, similar to that used for the RF regression model. In this matrix, one column represented the target variable (the number of PRRSV-positive submissions), while the remaining 12 columns contained lagged values of the target variable, corresponding to a time-delay embedding with a 12-week lag. Each row of the matrix was treated as a single input sequence to the RNN layer.

Building and training RNN is a complex task and involves careful tuning of multiple hyperparameters, including the optimal number of layers, the optimal number of nodes in each layer, the activation function, and the optimization algorithm. For model development, we employed *Keras*, an open-source, high-level neural networks Application Programming Interface (API). The *Keras* interface was accessed through R using the *keras* package, which was installed from its GitHub repository via the *devtools* package [28,29]. This interface requires the *TensorFlow* and *reticulate* packages, which were also installed using *devtools* [29–31]. Prior to model training, the input time-series data were reshaped to match the format required by Keras and its underlying TensorFlow back-end. Specifically, the input was structured to have a single feature dimension, as the dataset comprised only one univariate time series.

The RNN architecture was implemented using the `keras_model_sequential()` function to construct a linear stack of layers. The first layer was defined using one of the following functions: `layer_simple_rnn()`, `layer_gru()`, or `layer_lstm()`. The `units` parameter, which determines dimensionality of the output space, was varied during model tuning to optimize performance. The `input_shape` parameter was set to (12, 1), representing 12-time steps (lag order) and one variable per time step.

To enable uncertainty estimation during prediction, a second layer was added using the `layer_dropout()` function. While dropout is typically used during training for regularization – by randomly deactivating a fraction of the units to prevent overfitting – it is usually disabled during inference. However, to facilitate Monte Carlo (MC) dropout for generating prediction intervals, dropout was enabled during both training and prediction phases. The `rate` parameter, which determined the fractions of input units set to zero, was tuned to balance model performance and predictive uncertainty.

The third layer was created using the `layer_dense()` function to apply a linear transformation to the outputs of the previous layer. This layer was followed by the rectified linear unit (ReLu) activation function, commonly recommended for time-series forecasting tasks [32], as it introduces non-linearity and enables the model to capture complex patterns. The number of units in this dense layer was also tuned to enhance performance.

A fourth layer was added for additional regularization using another `layer_dropout()` function, with the optimized dropout rate during training.

Finally, the output layer was constructed using the `layer_dense()` function, designed to produce 104 output values based on the transformed features from the preceding layer.

For computational efficiency, the `callback_early_stopping()` function was used to terminate training when the monitored quantity ceased to improve, thereby reducing computational time and helping prevent overfitting.

The model's learning process was guided by the `compile` function, which specifies the loss function, optimization algorithm, and performance metrics. The mean squared error (MSE) was selected as the loss function to quantify the difference between the predicted and actual target values. The Adaptive Moment estimation (Adam) optimizer was employed to update model weights, offering fast and stable convergence. The model performance was monitored using both mean absolute error (MAE) and MSE metrics.

The model was trained on the time-series data using the fit() function, with optimized values for hyperparameters, including number of units, dropout rate, number of epochs, batch size, learning rate, and validation split through a systematic search process within the keras framework. The units parameter determines the dimensionality of the output space,

while dropout rate specifies the fraction of input units randomly set to zero to prevent overfitting. The epochs parameter defines the number of complete passes through the training set. The batch size parameter specifies the number of samples processed before the model's internal parameters are updated. The learning rate parameter controls the magnitude of weight updates during optimization, and the validation_split determines the proportion of training data reserved for evaluating the model's performance on a validation set during training.

## Evaluation of time-series models

To evaluate the forecasting performance of the time-series models, two validation approaches were employed. In each approach, the times series of weekly PRRSV-positive submissions was partitioned into training and test datasets.

In the first approach, the time series was partitioned into training and test datasets, comprising the first 85% and the final 15% of consecutive weekly observations. The training dataset, spanning January 2011 to July 2021, was used to develop a model (S1 Fig). The test dataset, covering August 2021 to July 2023, was used to evaluate the model's predictive performance. The model accuracy was assessed using standard predictive error metrics, which are described in the section below.

The second approach involved a simulated prospective analysis using a forecasting rolling origin technique. As in the previous method, the time series was divided into training and test datasets. The simulations began by training a model on the first 416 weeks, spanning January 3, 2011, to December 2018 (S2 Fig).

Subsequently, the model was retrained iteratively by extending the training dataset by one additional week at each step and generating forecasts for the following 12 weeks. The initial test dataset comprised the 12 weeks beginning January 7, 2019, which was used to evaluate the predictive accuracy.

In each iteration, the 12-week prediction window was advanced by one week. The model was retrained using the updated training data, and forecasts were generated for the new 12-week period. For each iteration, the mean and median of both the observed and predicted number of positive submissions were recorded. These values were later used to visualize the model's forecasting performance.

Additionally, predictive error metrics were calculated and recorded for each iteration. At the end of the simulation, the recorded mean, median, and performance measures across all iterations were used to compute the overall model's predictive accuracies, including the mean, Standard Deviation (SD), and Standard Error (SE).

## Assessment measures

To evaluate the model's forecasting performance in both validation methods, we employed commonly used error metrics: Root Means Squared Error (RMSE), Mean Absolute Error (MAE), Mean Percentage Error (MPE), and Mean Absolute Percentage Error (MAPE). Each metric captures different aspects of predictive accuracy.

RMSE and MAE are scale-dependent measures. RMSE penalizes large errors due to the squared term, while MAE provides a straightforward measure of the average magnitude of errors, regardless of direction. Minimizing RMSE tends to favor predictions close to the mean, whereas minimizing MAE yields predictions closer to the median.

In contrast, MPE and MAPE are unit-free and more interpretable across different scales. MPE measures systematic bias, indicating whether a model consistently overestimates (positive MPE) or underestimates (negative PME) the observed values. MAPE quantifies the average magnitude of percentage errors, independent of direction, offering a general measure of predictive accuracy.

Although no universal standard exists, MAPE is often interpreted using the following benchmarks in forecasting literature: < 10% indicates highly accurate forecasts; 10–20% suggests good forecasts; 20–50% reflects reasonable forecasts; and > 50% indicates poor accuracy [33,34].

Further details about each performance metric are available in the Supporting Information (S1 File).

## Results

### Descriptive statistics

The weekly time series of tested and PRRSV-positive submissions included 654 observations. Weekly submission counts ranged from 0 to 99. The average number of tested submissions per week increased from 30 in 2011–64 in 2023, suggesting an upward trend over the study period. The submission count time series is presented in Fig 1.

The number of PRRSV-positive submissions per week ranged from 0 to 23, with the average weekly count increasing from approximately 6 in 2011–13 in 2023. Zero counts, originating from the submission process (e.g., weeks with no submissions) comprised 1.2% of the data. Weekly positive submission counts were relatively evenly distributed across weeks and years, with no consistent seasonal pattern observed visually (Fig 1, S3 Fig). However, greater variability and occasional peaks in positive counts were more common during the winter and spring seasons (weeks 1–23 and 48–52), while the late summer and fall period (weeks 25, 29, 33 and 44) generally showed lower variability with fewer extreme values, except for substantial increases observed around weeks 25, 29, 33 and 44. Additionally, a gradual year-to-year increase in the number of weekly PRRSV-positive submissions was observed. In 2011, weekly positive counts ranged from 2 to 10, whereas in 2023, this range expanded from approximately 10 to over 20.

Seasonal decomposition of the time series is presented in Fig 2. A visual inspection reveals a short-term decline in positive counts between 2011–2014, followed by a gradual upward trend. This pattern suggests non-stationarity due to a changing mean over time. The amplitude of seasonal fluctuations appears to be weak, while the remainder shows substantial irregular variation.

Graphical and formal assessment of stationarity indicated the presence of a trend (S4 Fig). Results from the ADF unit root test supports this finding (tau$_3$ = −7.6, phi$_2$ = 9.5, phi$_3$ = 9.3, p-value < 0.001).

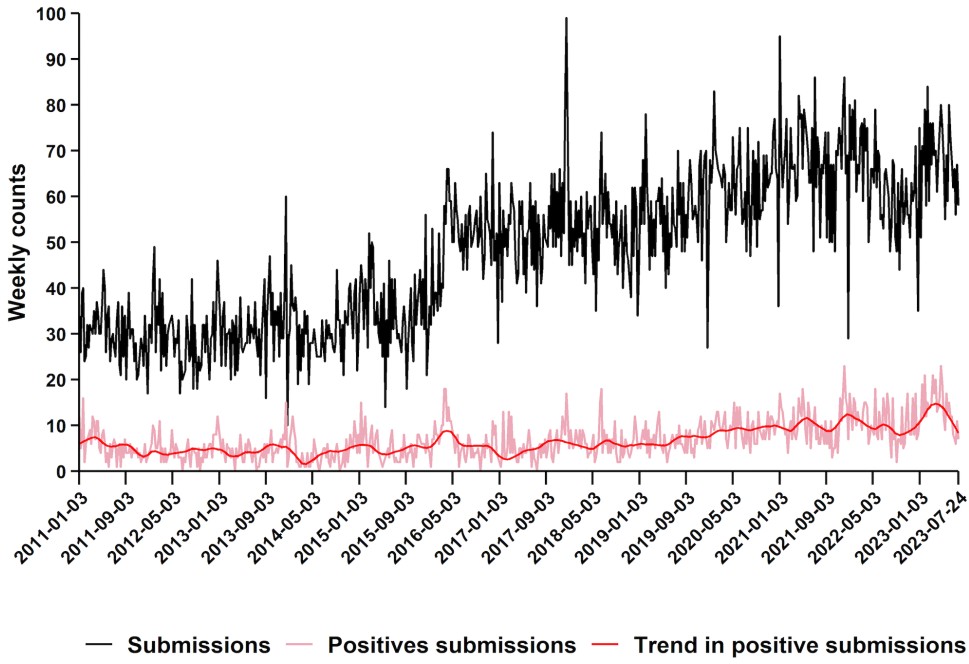

**Fig 1. Weekly PRRSV RT-PCR submissions and positive cases in Ontario (2011–2023).** Weekly counts of PRRSV RT-PCR and confirmed positive cases were obtained from the Animal Health Laboratory in Ontario from January 2011 to July 2023. The black line represents total weekly submissions, while the red line shows the number of confirmed positive cases. A slight upward trend (dark red) and sudden increases and drops as well as irregular structural patterns can be observed.

## Model validation through 104-week-ahead time-series forecasting

The time series of PRRSV-positive submissions were modeled with ARIMA, ETS, RF and RNN time-series models. The models were developed based on 550 observations, and the 104-week-ahead forecasting was used to validate the established models. Positive counts in the training dataset ranged from 0 to 18 (mean = 6.1, median = 6, variance = 12.3, SD = 3.5, SE = 0.2), and in the test dataset ranged from 2 to 23 (mean = 11.4, median = 11, variance = 21.5, SD = 4.6, SE = 0.5).

The predicted PRRSV positive counts with the ARIMA, ETS, RF and RNN time-series models are displayed in Fig 3. The predictive accuracy of the established time-series models is summarized in Table 1. Forecast deviations produced by the four time-series procedures are presented in S5 Fig.

Overall, the RMSE values range from 4.6 to 5.3, which is comparable to SD of the observed values. The MAE values range from 3.8 to 4.1, corresponding to approximately 19−20% of the full range of the test data and lower than the SD of the observed values. The slightly lower MAE compared to the RMSE suggests the presence of some outliers in the observed data. Together, the RMSE and MAE indicate moderately accurate model performance (Table 1, S5 Fig), especially when considered in the context of data range and variability. The MPE values range from −11.2% to 12.4%, reflecting a tendency of some models to underpredict and others to overpredict. The MAPE values, ranging from 36.4% to 41.7%, suggest that the forecasts are reasonably accurate.

In fitting ARIMA model, the data were square-root transformed to stabilize the variance. An autoregressive integrated moving average of order (2,1,1), ARIMA (2,1,1), was identified as the most parsimonious among the candidate models. First-order differencing effectively rendered the time-series stationary, indicating an underlying trend. Residual diagnostics

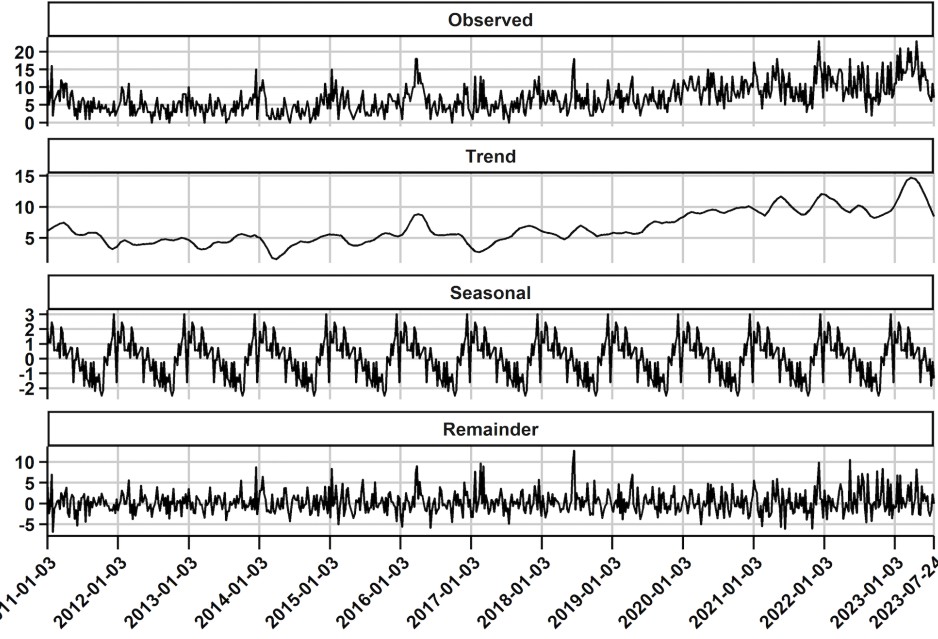

**Fig 2. Seasonal decomposition of weekly PRRSV-positive submissions.** The top panel displays the observed time series of weekly PRRSV-positive submissions. The second panel presents the estimated long-term trend component. The third panel illustrates the seasonal pattern, while the bottom panel shows the residuals – representing variations not explained by trend and seasonality. The actual counts are shown in black. Summary statistics for the observed data: min = 2, max = 23, mean = 11.4, median = 11; variance = 21.5. Variance decomposition: the trend component accounts for 39% of the total variance, indicating a dominant contribution; the seasonal component explains 9% of the variance, suggesting a weak repeating pattern; and the remainder accounts for 40% of the variance, indicating substantial irregular variation.

supported the adequacy of the ARIMA (2,1,1) model; however, visual inspection indicated that the model did not fully capture the underlying dynamics of the time series (Fig 3). Moreover, its predictive performance – evaluated using standard predictive error metrics – was suboptimal (Table 1).

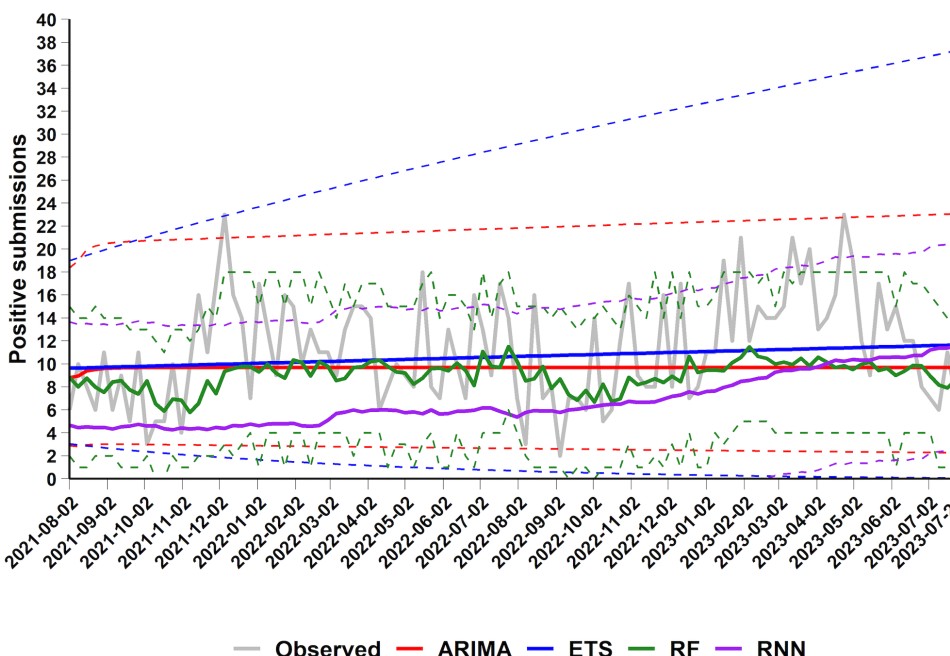

— Observed — ARIMA — ETS — RF — RNN

**Fig 3. Observed and predicted number of PRRSV-positive submissions (August 2021-July 2023) using a 104-week-ahead forecast approach with 95% prediction intervals.** Observed number of PRRSV-positive submissions in the test dataset (grey; min = 2, max = 23, mean = 11.4, median = 11, variance = 21.5, standard error = 0.5) alongside time-series predictions from ARIMA (red), ETS (blue), RF (green); and RNN (purple) models. Dashed lines represent 95% prediction intervals for each model; negative lower bounds are truncated to 0.

**Table 1. Predictive accuracy of the time-series models. The number of PRRSV-positive submissions was divided into training and test datasets. The training dataset included the first 550 weeks (January 2011–July 2021), and the test dataset comprised the subsequent 104 weeks (August 2021–July 2023). A 104-step-ahead forecast (N = 104) was applied to evaluate the predictive performance of ARIMA, ETS, RF, and RNN models.**

| Models[a,b] | N | Mean[a] | Median[a] | RMSE[c] | MAE[d] | MPE[e] | MAPE[f] |
|---|---|---|---|---|---|---|---|
| ARIMA (2,1,1) | 104 | 9.7 | 9.7 | 4.9 | 4.0 | 5.5% | 40.8% |
| ETS (A, N, N) | 104 | 10.4 | 10.4 | 4.6 | 3.8 | 12.4% | 41.7% |
| RF | 104 | 8.9 | 9.1 | 5.0 | 3.9 | −4.4% | 36.5% |
| RNN | 104 | 8.3 | 8.8 | 5.3 | 4.1 | −11.2% | 36.4% |

[a]Summary statistics of weekly PRRSV-positive submissions in the training dataset: min = 0, max = 18, mean = 6.1, median = 6, variance = 12.3, standard deviation (SD) = 3.5, standard error (SE) = 0.2

[b]Summary statistics of weekly PRRSV-positive submissions in the test dataset: min = 2, max = 23, mean = 11.4, median = 11, variance = 21.5, SD = 4.6, SE = 0.5

[c]Root Means Squared Error (RMSE)

[d]Mean Absolute Error (MAE)

[e]Mean Percentage Error (MPE)

[f]Mean Absolute Percentage Error (MAPE)

Based on the obtained error metrics, the model exhibits a slight overestimation bias and moderate relative errors (Table 1). The RMSE and MAE are 4.9 and 4.0, respectively, indicating the presence of some large individual errors. The MAE suggests that predictions deviate by an average of 4 units, which corresponds to approximately 19% of the total range of the test data. The positive MPE value (5.5%) indicates a tendency toward overprediction by 5.5% on average (Table 1, Fig 3, S5 Fig). The MAPE is 40.8%, suggesting that, on average, the predicted values deviate by approximately 41% from the observed values (Table 1, Fig 3, S5 Fig), which reflects moderate predictive accuracy.

In the ETS analysis, models with multiplicative errors were not considered as the time series contained zero values, which otherwise would lead to numerically unstable models. The square-root variance-stabilizing transformation was applied to the data. A simple exponential smoothing with additive errors model, ETS (A, N, N), was determined to be the most appropriate model based on Akaike's (AIC), AIC corrected (AICc), and Bayesian (BIC) information criteria. The residual analysis supported the selected model.

The RMSE and MAE are 4.6 and 3.8, respectively – the lowest among all evaluated time-series models. This suggests that ETS (A, N, N) achieves superior predictive accuracy; however, the relatively high MAE error rate, which is approximately 18% of the total range of the test data, may indicate the presence of outliers (Table 1). Additionally, visual inspection of the predictions (Fig 3) reveals that the model appears insensitive to fluctuations in the observed values. This observation is further supported by the fact that the MPE and MAPE are 12.4% and 41.7%, respectively – the largest values among four evaluated time-series models. It is important to note that the large positive MPE (12.4%) indicates that the ETS (A, N, N) model tends to overpredict on average by 12.4% (Fig 3, S5 Fig). Additionally, the relatively large MAPE value (41.7%) suggests that, on average, the predictions deviate from the actual values by approximately 42%, indicating that the model performance is reasonable, but not particularly strong.

For the RF regression model, the optimal configuration consisted of 500 trees with two variables considered at each split. As shown in Fig 3, this model exhibited greater sensitivity to fluctuations in the observed positive counts compared to the ARIMA (2,1,1) and ETS (A, N, N) models.

The RMSE value is 5 – higher than for the ARIMA (2,1,1) and ETS (A, N, N) models (Table 1). The MAE is 3.9, indicating a moderate average error that is approximately 19% of the total range of the test data. This means that, on average, the predictions deviate from the actual values by 3.9 units. The slightly lower MAE compared to the RMSE suggests the presence of occasional large prediction errors. The MPE of −4.4% is the lowest (in absolute terms) among all evaluated time-series models (Table 1), indicating that this model tends to underpredict slightly on average (Fig 3, S5 Fig). The MAPE value of 36.5% is also the lowest compared to the ARIMA (2,1,1) and ETS (A, N, N) models (40.8% and 41.7%, respectively), suggesting that, on average, the model predictions deviate from the observed values by about 37%. This supports that the RF model performs reasonably well.

The best performing RNN regression model employed an LSTM architecture with 64 units, a dropout rate of 0.2 applied in the dropout layer, and a dense layer comprising 128 units. The predictive performance of the RNN was comparable to that of the RF regression model. The RNN effectively captured weekly fluctuations – both increases and decreases – in PRRSV-positive submissions (Fig 3, S5 Fig). Predictive accuracy metrics were similar between two models (Table 1). However, the MPE for the RNN is −11.2%, indicating an average underprediction of approximately 11%. This underestimation bias is slightly more pronounced than in the RF model.

### Validation using rolling 12-step-ahead forecast approach

We evaluated the forecasting performance of four time-series models – ARIMA, ETS, RF, and RNN – using a rolling-origin evaluation with a 12-week forecast horizon and a non-fixed training window, to predict the number of PRRSV-positive submissions over time. The predicted mean PRRSV-positive counts are displayed in Fig 4. The predictive accuracy of the time-series models is summarized in Table 2.

 

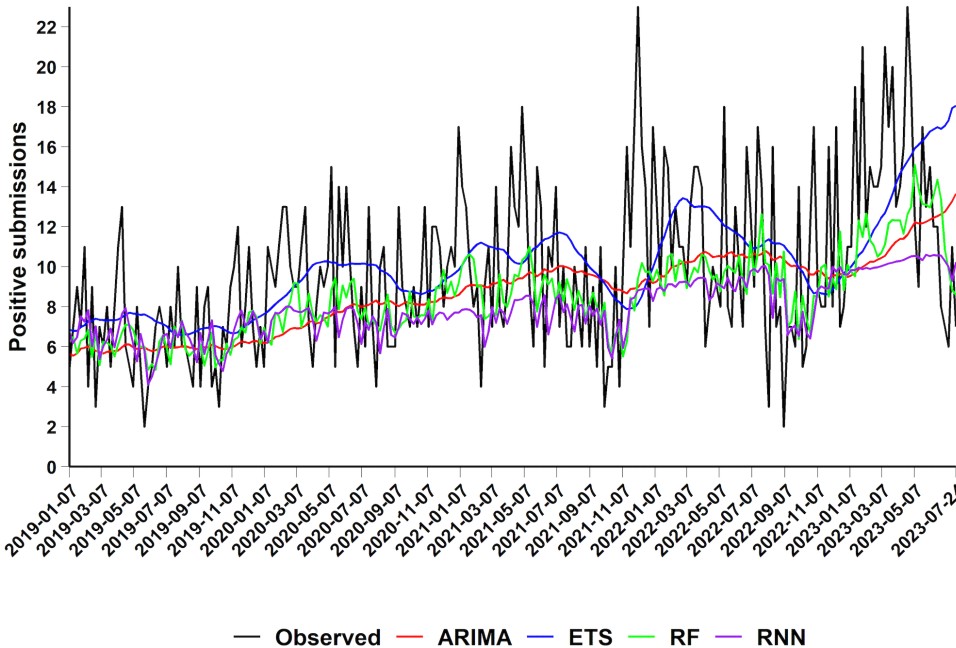

**Fig 4. Observed and predicted mean number of PRRSV-positive submissions (January 2019–July 2023) using a 12-step-ahead rolling forecast approach.** Observed number of PRRSV-positive submissions in the test dataset (black), overlaid with time-series predictions from ARIMA (red), ETS (blue), RF (green); and RNN (purple) models.

Among all models, the RF and RNN time-series models showed greater sensitivity to short term fluctuations in the data, capturing both increases and decreases in case counts.

In terms of predictive accuracy, all four models performed comparably, as indicated by similar RMSE and MAE values (Table 2). However, the RF model consistently achieved the lowest RMSE (3.79) and MAE (3.11), suggesting better over-all predictive performance.

**Table 2. Predictive accuracy of the time-series models. The number of PRRSV-positive submissions was partitioned into training and test datasets. The training dataset included the first 416 weeks (January 2011–December 2018), and the test dataset comprised 12 weeks starting January 7 to March 25, 2019. A rolling 12-step-ahead forecast was applied to assess the predictive performance of ARIMA, ETS, RF, and RNN models.**

| Models | N[a] | Estimated mean±SE[b] | Estimated median±SE[b] | RMSE[c] | | | MAE[d] | | | MPE[e] | | | MAPE[f] | | |
|---|---|---|---|---|---|---|---|---|---|---|---|---|---|---|---|
| | | | | Mean | SD[g] | SE | Mean | SD[g] | SE | Mean% | SD[g] | SE | Mean% | SD[g] | SE |
| ARIMA | 227 | 8.7±0.1 | 8.6±0.1 | 3.84 | 1.23 | 0.08 | 3.20 | 1.08 | 0.07 | 2.93% | 26.19 | 1.74 | 36.23% | 15.71 | 1.04 |
| ETS | 227 | 10±0.2 | 10±0.2 | 3.81 | 1.29 | 0.08 | 3.19 | 1.13 | 0.08 | 18.21% | 30.04 | 1.99 | 40.48% | 19.20 | 1.33 |
| RF | 227 | 8.5±0.1 | 8.5±0.1 | 3.79 | 1.15 | 0.07 | 3.11 | 0.93 | 0.06 | 0.36% | 18.01 | 1.19 | 34.29% | 12.35 | 0.82 |
| RNN | 227 | 7.9±0.1 | 8±0.1 | 4.02 | 1.35 | 0.09 | 3.32 | 1.13 | 0.08 | −5.89% | 18.34 | 1.22 | 34.68% | 10.66 | 0.71 |

[a]Number of weeks in the test dataset (N)

[b]Summary statistics of weekly PRRSV-positive submissions: Mean±standard error (SE): 9.9±0.2, median±standard error (SE): 9.5±0.2.

[c]Root Means Squared Error (RMSE)

[d]Mean Absolute Error (MAE)

[e]Mean Percentage Error (MPE)

[f]Mean Absolute Percentage Error (MAPE)

[g]Standard Deviation (SD)

The MPE values suggest that the ARIMA (2.93%), ETS (18.21%), and RF (0.36%) models tend to overestimate the counts on average, whereas the RNN model exhibits a slight underestimate bias (−5.89%). Notably, the RF produced the smallest absolute bias, with the MPE value close to zero (0.36%). This may be attributed to substantial errors occurring in both positive and negative directions, which effectively offset each other in the MPE calculation.

The MAPE values across the models are relatively high, ranging from 34.29% (RF) to 40.48% (ETS), reflecting moderate forecasting error relative to the actual values. Despite its low bias, the RF's MAPE value indicates substantial prediction errors. Although all MAPE values are above the 10% thresholds for "high accuracy", the RF still outperformed other models on this metric, reinforcing its overall superiority in both error magnitude and bias over other approaches considered herein.

Overall, the RF model demonstrated the highest forecast accuracy across all four predictive performance criteria, outperforming the ARIMA, ETS and RNN models.

## Discussion

Published research exploring alternative methods for analyzing time series of PRRSV in swine population for surveillance purposes remains limited. To the best of our knowledge, this is the first study to model and predict PRRSV frequency using diverse time-series forecasting techniques. We evaluated two statistical models – ARIMA and ETS; a RF time-series regression model representing machine learning approaches; and a RNN from the domain of deep learning. Each of these modeling approaches has distinct strengths and limitations in capturing temporal patterns in weekly PRRSV-positive submissions.

ARIMA and ETS models are commonly used in health-related time-series forecasting due to their simplicity and capacity to model trends and seasonality. More recently, RF and RNNs have been applied, offering greater flexibility in capturing complex and nonlinear patterns. Thus, in this study we integrated these techniques, leveraging their respective strengths. Using laboratory data on PRRSV submissions from the AHL, we found that the RF regression model outperformed ARIMA, ETS, and RNN models in forecasting accuracy, suggesting its greater suitability for modeling the temporal dynamics of PRRSV occurrence.

Descriptive analyses and graphical assessments indicated large week-to-week fluctuations in the number of positive results across all seasons. The few zero counts were observed in some weeks, and they likely reflected variability in the submission process rather than a true absence of disease. Given their small proportion, these zero values did not affect the time series models (e.g., after differencing with ARIMA, the distribution of residuals appeared reasonably symmetric). The high variability of test results from samples submitted throughout the years have obscured any clear signal in the data, which is likely why the ARIMA and ETS models did not detect a pronounced seasonal pattern. The fact that no clear seasonal pattern was identified in our analysis contrasts with the established epidemiological characteristics of PRRSV, which typically exhibits increased transmission during colder months, as well as with findings from previous studies on seasonality [35–37]. Several factors may explain this discrepancy. One possible explanation is the influence of frequent testing conducted across different phases of disease investigation, management, intervention, or monitoring, both during and after herd closure. As an example, previous work has shown that a single premises may submit one or multiple laboratory submissions within a given week [7], with a median of one submission (IQR = 0) and up to 13 submissions per week in extreme cases. Similarly, our dataset included submissions classified as either diagnostic or monitoring. In theory, restricting the analysis to diagnostic submissions might have improved forecasting accuracy. However, we chose not to pursue this approach for several reasons.

In explanatory analyses, we applied time-series modeling separately to datasets containing (i) only diagnostic submissions, and (ii) both diagnostic and monitoring submissions to assess whether model accuracy differed substantially

between them. The result indicated that using only diagnostic submissions occasionally led to slight improvements in model accuracy based on the RMSE and MAE metrics (but not based on the MPE and MAPE metrics, which were higher). However, these differences were marginal and inconsistent across evaluation approaches.

Previous work has suggested that the reasons for submissions (i.e., diagnostic vs. monitoring) may have been applied differently or inconsistently over the duration of this study [7], potentially introducing an additional source of bias or uncertainty. Therefore, we opted to include both diagnostic and monitoring data in our final modeling.

Finally, it is also possible that seasonal transmission patterns vary by geographic region and are influenced by local climate conditions, changes in farm management, or biosecurity measures over time [36]. Nonetheless, general climatic conditions and swine production practices, including biosecurity, in regions where PRRSV seasonality has been formally assessed do not appear differ substantially from those in the preset study area.

It appears that a temporal pattern remained stable in the data as the four time-series models were able to capture an increasing temporal trend. This finding could be a consequence of the emergence of new viral strains that evade existing immunity; some gaps in biosecurity management; improved diagnostic and laboratory reporting systems; improved laboratory methods; higher testing rates or other factors.

In this study, we employed two distinct approaches to evaluate predictive performance of ARIMA, ETS, RF and RNN times-series models, and also to compare the potential of the RNN – despite the relatively small dataset – to capture nonlinear and complex dependencies that may not adequately modeled by traditional statistical techniques. The results indicated that some models were more sensitive to noise than others. We acknowledge that the overall forecasting performance of these models was moderate (e.g., MAPE ~ 35–42%), which likely reflects high variability and sporadic nature of PRRSV submissions, including periods with zero activity and diverse submission types.

Both ARIMA and ETS models failed to adequately capture the time-series dynamics in either validation approaches. These models tended to overestimate forecasts, and their error metrics were consistently higher than those obtained from the RF and RNN models. The observed overestimation was likely attributable to the inherent model limitations [19,20,38]. First, high variability across seasons masked any clear seasonal pattern, reducing models' ability to accurately capture temporal patterns. Second, although the data were square-root transformed to stabilize the variance, this transformation could not fully account for occasional large spikes or irregular fluctuations, and back-transforming predictions may have amplified overestimation. Third, low values in the data likely contributed to overestimation, as the models tended to expect higher values based on preceding peaks. Finally, the relatively simple ARIMA and ETS structures are limited in their abilities to capture complex, non-linear patterns inherent in weekly PRRSV data.

The RF and RNN models adapted reasonably well to the abrupt changes observed in the PRRSV time series, although both tended to underestimate the forecasted values. Overall, these models outperformed the fitted ARIMA and ETS in predicting PRRSV frequency, with the RF demonstrating slightly higher accuracy. These findings highlight the versatility and predictive effectiveness of RF and RNN approaches for modeling complex epidemiological time-series data.

However, their limitations should be carefully considered. Both RF and RNN are often regarded as "black box" models due to their limited interpretability compared to more transparent methods such as ARIMA and ETS. Moreover, the time-series dataset – comprising 654 weekly observations – was relatively small, which posed challenges for effectively training the RNN. To mitigate the risk of overfitting associated with limited data, several regularization strategies were applied, including dropout, recurrent dropout, and early stopping. In addition, windowing technique (e.g., using overlapping 12-week input windows) was employed to increase the number of training samples and enable the model to learn from a broader range of temporal patterns.

Implementing the rolling 12-week-ahead forecasting approach using RF and RN also presented practical challenges. The validation process was computationally intensive due to model complexity, high memory requirements, and the difficulty of identifying optimal configurations.

Several practical implications can be drawn from this research. Using a combination of different forecasting techniques is advantageous when predicting phenomena characterized by complex and nonlinear dynamics. The weekly PRRSV frequencies did not show clear seasonal patterns. A significant increasing trend was identified in positive counts.

While additional algorithms or further fine-tuning of the methods used in this study may offer some improvements, the time series of weekly PRRSV-submissions contained substantial, irregularly spaced outliers, with error remaining a dominant component. This error could potentially have been reduced by focusing solely on diagnostic data; however, excluding monitoring submissions provided only minor accuracy gains while potentially introducing bias and uncertainty due to inconsistencies in how submission reasons were interpreted. Enhancing the accuracy and consistency of submission information – through improved training and communication with diagnostic personnel – may improve the epidemiological quality of PRRSV surveillance data.

Overall, the ARIMA and ETS models tended to overestimate predicted PRRSV frequencies, whereas the RF and RNN models tended to underestimate them. Despite moderate prediction errors, all four models effectively captured key temporal trend, and some provided early indications of increased activity, supporting their potential utility for predictive purposes in surveillance. Among the evaluated methods, the RF model demonstrated the highest predictive accuracy. However, its current level of accuracy, as evaluated by the MAPE metrics, remains insufficient for implementation in regular interactive animal pathogen dashboards (IAPD). Further refinement of the model and/or processing of input data is needed to improve performance and provide clearer indications of incident cases submitted for diagnostic purposes.

Future research should focus on enhancing forecasting efficacy by incorporating additional covariates – such as farm-level characteristics (e.g., herd size, geographical location, and local climatic conditions) – that were unavailable in the current dataset but may influence disease occurrence. Moreover, improving the precision and standardization of submission records, exploring temporal aggregation (e.g., monthly data), expanding the temporal coverage of the dataset, or employing ensemble or hybrid modeling approaches that combine statistical and deep learning methods may further strengthen predictive performance. Such efforts could help determine whether the observed performance levels are sufficient for operational use within the regular PRRSV surveillance system.

The findings of this study are most directly applicable to Ontario, Canada, reflecting local management practices, biosecurity standards, and surveillance participation. Cautions should be exercised when extrapolating these results to other provinces or countries with differing production systems or epidemiological contexts. Nonetheless, the time series modeling framework applied here provides a transferable approach that can be adapted to other regions to explore local PRRSV dynamics and support data-driven surveillance.

## Supporting information

**S1 Fig. Partitioning of the weekly time series of PRRSV-positive submission (January 2011-July 2023) for a 104-week-ahead forecasting approach.** The green color indicates the training period (January 2011- July 2021), and the red color represents the testing period (August 2021 – July 2023).
(TIF)

**S2 Fig. Partitioning of the weekly time series of PRRSV-positive submissions using a 12-week-ahead rolling forecasting approach.** The green color indicates the training period in each iteration, and the red color represents the testing period.
(TIF)

**S3 Fig. Distribution of weekly PRRSV-positive submissions in Ontario (2011–2023).** Boxplots for each week illustrate the distribution of weekly PRRSV-positive submissions across 13 years. The centre dark line in each boxplot shows the median, and the box edges represent the first and third quartiles (interquartile range). Whiskers extend to the most extreme values within 1.5 times of the interquartile range, while points beyond this range (circles) indicate outliers. Winter

and spring (weeks 1–23 and 50–52) exhibit broader interquartile ranges and more outliers, indicating higher variability and occasional surges in case counts. Late summer and fall (weeks 33–50) show lower variability and fewer extreme values. (TIF)

**S4 Fig. Assessment of stationarity in the weekly time series of PRRSV-positive submissions using autocorrelation and partial autocorrelation plots.** (A) The autocorrelation function (ACF) shows a gradual decay across increasing lags, consistent with non-stationarity due to trend or seasonality. (B) The partial autocorrelation function (Partial ACF) displays a significant spike at lag 1, followed by smaller spikes at subsequent lags. This pattern suggests non-stationarity behavior and supports the presence of autoregressive structure in the time series. (TIF)

**S5 Fig. Forecast errors for weekly PRRSV-positive submissions (August 2021 to July 2023) using a 104-week-ahead forecast approach.** Forecast errors between predicted and observed weekly counts of PRRSV-positive submissions are shown for the period August 2021 to July 2023. Predictions were generated using four time-series models: autoregressive integrated moving average (ARIMA, red), exponential trend smoothing (ETS, blue), random forest (RF, green), and recurrent neural network (RNN, purple). The actual counts are shown in black. Summary statistics for the observed data include: min = 2, max = 23, mean = 11.4, median = 11; variance = 21.5, standard error = 0.5. (TIF)

**S6 Fig. Recurrent neural network.** A diagram of simple recurrent network (RNN). $X$ denotes the input vector, $HU$ represents the hidden units, and $Y$ is the output vector. $t$ indicates discrete time steps. $W_U$, $W_V$, and $W$ denote the trainable weight matrices of the RNN. Dashed lines indicate trainable connections. At each time step $t$, an input $X_t$ is processed to produce an output $Y_t$, with the hidden units maintaining a partial memory of previous input-output states. The memory updates over time to enhance the network's ability to model sequential dependencies. (TIF)

**S7 Fig. Elman recurrent neural network.** A diagram of Elman recurrent neural network. Dashed lines represent trainable connections. (TIF)

**S1 File. Extended description of time-series methods.** (DOCX)

## Acknowledgments

We would like to thank the members of the Ontario Animal Health Network swine group for their feedback and advice.

## Author contributions

**Conceptualization:** Tatiana Petukhova, Maria Spinato, Zvonimir Poljak.

**Data curation:** Tatiana Petukhova, Tanya Rossi, Pauline Nelson-Smikle.

**Formal analysis:** Tatiana Petukhova.

**Funding acquisition:** Maria Spinato, Zvonimir Poljak.

**Investigation:** Davor Ojkic.

**Methodology:** Tatiana Petukhova, Zvonimir Poljak.

**Project administration:** Zvonimir Poljak.

**Resources:** Zvonimir Poljak.

**Software:** Tatiana Petukhova, Pauline Nelson-Smikle.

**Supervision:** Maria Spinato, Zvonimir Poljak.

**Validation:** Maria Spinato, Zvonimir Poljak.

**Visualization:** Tatiana Petukhova.

**Writing – original draft:** Tatiana Petukhova.

**Writing – review & editing:** Maria Spinato, Tanya Rossi, Michele T. Guerin, Cathy A. Bauman, Pauline Nelson-Smikle, Davor Ojkic, Zvonimir Poljak.

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
