## [Decision Letter · Decision Letter 0]

24 Sep 2025

Dear Dr. Tatiana Petukhova,

Thank you for submitting your manuscript to PLOS ONE. After careful consideration, we feel that it has merit but does not fully meet PLOS ONE’s publication criteria as it currently stands. Therefore, we invite you to submit a revised version of the manuscript that addresses the points raised during the review process.

We look forward to receiving your revised manuscript.

Kind regards,

Ahmad Salimi

Academic Editor

PLOS ONE

Journal Requirements:

“The author who received funding: ZP

Grant number: 499163

This project was funded by the University of Guelph's Food From Thought program

The author who received funding: MS

Grant number: 499170

This project was funded by the University of Guelph's Food From Thought program”

“This project was funded by the University of Guelph’s Food From Thought program. We would like to thank the members of the Ontario Animal Health Network swine group for their feedback and advice.”

“The author who received funding: ZP

Grant number: 499163

This project was funded by the University of Guelph's Food From Thought program

The author who received funding: MS

Grant number: 499170

This project was funded by the University of Guelph's Food From Thought program”

Reviewers' comments:

Reviewer's Responses to Questions

**Comments to the Author**

1. Is the manuscript technically sound, and do the data support the conclusions?

Reviewer #1: Partly

Reviewer #2: Partly

2. Has the statistical analysis been performed appropriately and rigorously?

Reviewer #1: N/A

Reviewer #2: Yes

3. Have the authors made all data underlying the findings in their manuscript fully available?

Reviewer #1: No

Reviewer #2: Yes

4. Is the manuscript presented in an intelligible fashion and written in standard English?

Reviewer #1: Yes

Reviewer #2: Yes

Reviewer #1: ‏The study is valuable; however, clarification on the following points would strengthen the manuscript

1. Data and Scope

• Could you clarify why both diagnostic and monitoring submissions were included? Would separating them improve model accuracy?

• Were farm-specific factors (e.g., size, location, climate) considered in the analysis?

2. Models

• How were hyperparameters optimized for the RF and RNN models? Was a systematic tuning procedure applied?

• Did you explore transformations (e.g., log or power) for the count data to improve ARIMA model performance?

3. Interpretation of Results

• Given the relatively high MAPE (~35–42%), what are the practical implications for PRRSV surveillance?

• Why was no clear seasonal pattern observed, while previous studies indicate higher transmission in colder months?

4. Limitations and Applicability

• Do the authors believe the results are generalizable to other provinces or countries with different conditions?

• Have ensemble or hybrid modeling approaches been considered to potentially enhance predictive accuracy?

Potential Weaknesses / Issues

1. Methodology

• Limited explanation of submission type selection and its potential impact on model outcomes.

• Insufficient details on hyperparameter optimization for RF and RNN models.

• No discussion of handling zero values in count data for ARIMA modeling.

2. Prediction Accuracy

• High MAPE values (35–42%) indicate moderate predictive performance; this should be discussed in terms of practical surveillance applications.

• ARIMA and ETS models tend to overpredict; reasons for these biases are not fully explained.

3. Results Interpretation

• Lack of clear seasonal patterns contradicts prior epidemiological knowledge; explanations are speculative and not deeply analyzed.

• Some discussion sections are lengthy and could be clarified for better readability.

4. Generalizability

• Data are limited to one province; environmental and management factors are not fully incorporated, limiting broader applicability.

Reviewer #2: 1. The rationale for selecting only ARIMA, ETS, RF, and RNN is unclear. The authors should clarify why more advanced or hybrid methods (e.g., Prophet, XGBoost, or ensemble models) were not considered.

2. The reported forecasting error (MAPE ≈ 35%) is relatively high. Conclusions regarding the suitability of RF as the “best model” should be tempered to reflect this limitation.

3. The study did not detect seasonal patterns, which contradicts previous reports. Stronger statistical tests or clearer scientific explanations are required to justify this discrepancy.

4. The combination of diagnostic and monitoring submissions may have introduced noise. A sensitivity analysis restricted to diagnostic data would strengthen the study.

5. Given the relatively small dataset, the use of RNN may not be appropriate. This limitation should be more clearly discussed.

Overall, the manuscript is promising, but revisions are needed to align conclusions with the data and methodological limitations.

**Do you want your identity to be public for this peer review?** For information about this choice, including consent withdrawal, please see our Privacy Policy

Reviewer #1: No

Reviewer #2: No

---

## [Author Response · Author response to Decision Letter 1]

6 Nov 2025

Editor’s comments:

Author response:

We have reviewed the PLOS ONE’s style requirements and ensured our manuscript fully complies with them, including file naming and formatting guidelines.

Please note that PLOS One has specific guidelines on code sharing for submissions in which author-generated code underpins the findings in the manuscript. In these cases, we expect all author-generated code to be made available without restrictions upon publication of the work. Please review our guidelines at https://journals.plos.org/plosone/s/materials-and-software-sharing#loc-sharing-code and ensure that your code is shared in a way that follows best practice and facilitates reproducibility and reuse.

Author response:

We have reviewed the PLOS ONE guidelines on code sharing and will ensure that our author-generated code is shared, if required, in accordance with these recommendations to promote reproducibility and reuse.

Thank you for stating the following financial disclosure. Please state what role the funders took in the study. If the funders had no role, please state: "The funders had no role in study design, data collection and analysis, decision to publish, or preparation of the manuscript."

Author response:

Please include the following Role of Funder statement:

We note that you have indicated that there are restrictions to data sharing for this study. PLOS only allows data to be available upon request if there are legal or ethical restrictions on sharing data publicly. For more information on unacceptable data access restrictions, please see http://journals.plos.org/plosone/s/data-availability#loc-unacceptable-data-access-restrictions.

Author response:

Data used for this study cannot be shared publicly due to our REB application. Part of the REB application is also founded on the confidentiality agreement between the principal investigator and the laboratory providing access to the data. They are confidential and, in some cases, could be identifiable even after de-identification. Contact information for our institutional REB committee is reb@uoguelph.ca.

Author response:

Please retain the current Data Availability statement. The data cannot be shared publicly due to confidentiality.

Thank you for stating the following in the Acknowledgments Section of your manuscript:

“This project was funded by the University of Guelph’s Food From Thought program. We would like to thank the members of the Ontario Animal Health Network swine group for their feedback and advice.”

“The author who received funding: ZP

Grant number: 499163

This project was funded by the University of Guelph's Food From Thought program

The author who received funding: MS

Grant number: 499170

This project was funded by the University of Guelph's Food From Thought program”

Author response:

All funding-related text has been removed from the Acknowledgements section and the main manuscript. We would like to change the current Funding Statement as the following:

This project was supported by the Food from Thought program at the University of Guelph, funded through the Canada First Research Excellence Fund (Grant number: 499163, the author who received funding: ZP; Grant number: 499170, The author who received funding: MS), and by the Ontario Agri-Food Innovation Alliance (grant number: 100022, the author who received funding: ZP).

Author response:

No such recommendations were made by the reviewers.

Reviewer #1

1. Data and Scope

Reviewer comment:

Could you clarify why both diagnostic and monitoring submissions were included? Would separating them improve model accuracy?

Author response:

Thank you for this valuable comment. We fully agree with the rationale for separating diagnostic and monitoring submissions to improve model accuracy. This approach was indeed part of initial consideration before conducting a more detailed data analysis. In our previous publication (Petukhova T. et al., Front Vet Sci. 2025; 12:1528422. doi:10.3389/FVETS.2025.1528422), we specifically investigated the impact of splitting diagnostic and monitoring data. The relevant findings from that study are quoted as follows “In addition, a pattern in the number of monitoring submissions submitted per month was observed, where three distinct periods were visible (Figure 1) with abrupt changes in the number of submissions that coincided with the time that the calendar year ends and starts. The period until 2017 was characterized by a progressively higher number of submissions over time (mean = 38.7, sd = 15.9), followed by a generally lower number of submissions between January 2017 and December 2021 (mean = 27.3 sd = 9.7), and a subsequent increase in the number of monitoring submissions starting in January 2022 (mean = 65, sd = 14.7) (Figure 1). Similar patterns were noted in monitoring submissions in both nursery (Supplementary Figure 1) and grower/finisher pigs (Supplementary Figure 2). An interesting pattern was also observed among submissions for suckling pigs: while the number of all submissions increased over time and particularly since the start of 2020 (Supplementary Figure 3), the number of monitoring submissions in this production class was low until 2022 (Supplementary Figure 4)”.

In the discussion of previously published paper, we specifically made a note about this observation by stating “Research into understanding how end users define a monitoring vs. a diagnostic submission, and whether they adhere to their definition when submitting samples for pathogen testing, is warranted before relying too much on its interpretation. Classification into monitoring and diagnostic submissions as the two basic categories is sensible, but it would be prudent to understand practices that contribute to defining these submission types, and motivations of people who submit samples to indicate certain submission type....”

In brief, we believed that by using all the data, we would be better at investigating any short- or long-term trends, as there were clearly factors that impacted such classification that we could not capture.

Nonetheless to address your comment fully, we applied time series modeling separately to datasets containing (i) only diagnostic submissions, and (ii) both diagnostic and monitoring submissions to assess whether model accuracy differed substantially between them. The result indicated that using only diagnostic submissions led to slight improvements in model accuracy based on RMSE and MAE metrics (but not based on the MPE and MAPE metrics, which were higher). However, these differences were marginal and inconsistent across evaluation approaches. For example, in the first evaluation approach (RF model), diagnostic-only data yielded RMSE = 4.3, MAE = 3.3, MPE = 7.9%, MAPE = 40.3%, whereas modeling both diagnostic and monitoring submissions resulted in RMSE = 5.0, MAE = 3.9, MPE = -4.4%, MAPE = 36.5%. In the second evaluation approach, the corresponding values were RMSE = 2.92, MAE = 2.91, MPE = 4.05%, MAPE = 36.5% (diagnostic-only) and RMSE = 3.79, MAE = 3.11, MPE = 0.36%, MAPE = 34.29% (combined data).

Thus, given that excluding monitoring submissions did not provide substantial gains with all the issues that we identified in the previous paper, and with potentially introducing bias and uncertainty due to inconsistent reporting of submission reasons, we opted to include both diagnostic and monitoring data in our final modeling. This decision is consistent with previous work demonstrating the value of integrating both submission types for comprehensive surveillance (Petukhova T. et al., Front Vet Sci. 2025; 12:1528422. doi:10.3389/FVETS.2025.1528422). We have clarified this point in the Discussion section of the revised manuscript (lines 582-596 and 650-658).

Reviewer comment:

Were farm-specific factors (e.g., size, location, climate) considered in the analysis?

Author response:

Farm-specific factors (e.g., size, location, climate) were not included in the analyses because this submission information was not available in the dataset. We acknowledge that these variables could potentially influence disease dynamics, and we plan to incorporate them in future work if such data become accessible. We add this information as a limitation in the manuscript (see lines 668-671 of the revised manuscript).

2. Models

Reviewer comment:

How were hyperparameters optimized for the RF and RNN models? Was a systematic tuning procedure applied?

Author response:

The optimal hyperparameters for the RF were identified using the tuneRF() function and subsequently applied in the RF regression procedure. For the RNN model, hyperparameter tuning was performed through a systematic search process within the keras framework. Model performance was evaluated using MAE and MSE metrics, and early stopping was applied to prevent overfitting. These clarifications have been added to the revised manuscript (lines 226-230 and 300-309).

Reviewer comment:

Did you explore transformations (e.g., log or power) for the count data to improve ARIMA model performance?

Author response:

In fitting the ARIMA model, we used the auto.arima() function with the Box-Cox transformation parameter set to “auto”. This allowed the function to automatically select an appropriate variance-stabilizing transformation based on the data, helping to improve ARIMA model performance. We have clarified this point in the Methods section (see lines 160-164 of the revised manuscript). We also reported which transformation was selected by the function in the Results section (line 450).

3. Interpretation of Results

Reviewer comment:

Given the relatively high MAPE (~35–42%), what are the practical implications for PRRSV surveillance?

Author response:

We acknowledge that the MAPE values of ~35-42% indicate moderate predictive error in our models. Nevertheless, all four models effectively captured key temporal trend, and some provided early indications of increasing activities, supporting their potential utility for predictive purposes in surveillance. However, the current level of accuracy remains insufficient for implementation in regular interactive animal pathogen dashboards (IAPD). Further refinement of the model and/or processing of input data is needed to improve performance and provide clearer indications of incident cases submitted for diagnostic purposes. The precision and standardization of submission records, aggregating data at a larger temporal scale (e.g., monthly) or incorporating additional farm-level variables may further improve predictive performance in future studies. We added this practical implication in the Discussion section of the revised manuscript (lines 660-676).

Reviewer comment:

Why was no clear seasonal pattern observed, while previous studies indicate higher transmission in colder months?

Author response:

In our study, the modeled data consisted of test results from samples submitted throughout the years, showing substantial variability across all seasons. Descriptive statistics and graphical assessments indicated large week-to-week fluctuations in the number of positive results during winter (e.g., 2-17 positives in week 1, 3-17 in week 48, 3- 23 in week 49), spring (e.g., 2-21 in week 11, 0-20 in week 13, 4-23 in week 17), summer (e.g., 1-18 in week 25, 0-17 in week 29, 3-16 in week 33), and fall (e.g., 2-17 in week 44). The variability in the positive counts also increased notably since 2017.

This high variability across seasons likely obscured any clear seasonal signal in the data. Several potential factors may have contributed to the variability, including frequent testing conducted across different phases of disease investigation, management, intervention, or monitoring, both during and after herd closure. It is also possible that seasonal transmission patterns vary by geographic region and are influenced by local climate conditions, changes in farm management, or biosecurity measures over time. As shown in Figure 2, the seasonal component explained only 9% of the variance, suggesting that seasonality was weak relative to other sources of variation, which likely explains why the ARIMA and ETS models did not detect a pronounced seasonal pattern. In the revised manuscript, we provided more details in the Result section and expanded the discussion to clarify possible reasons why seasonal patterns were not detected by the time-series models (lines 376-389, 572-580, and 597-601).

4. Limitations and Applicability

Reviewer comment:

Do the authors believe the results are generalizable to other provinces or countries with different conditions?

Author response:

We acknowledge that the results are primarily reflect the epidemiological and management conditions in Ontario, Canada, where the data were collected. Therefore, direct generalization to other provinces or countries should be done with caution, as differences in swine production systems, biosecurity practices, and surveillance intensity may influence PRRSV dynamics. Nevertheless, the analytical framework and modeling approaches presented in this study can be readily applied to similar datasets from other regions to assess local patterns and inform surveillance strategies under different conditions. We have clarified this point in the Discussion section of the revised manuscript (lines 677-682).

Reviewer comment:

Have ensemble or hybrid modeling approaches been considered to potentially enhance predictive accuracy?

Author response:

We appreciate the reviewer’s suggestion regarding ensemble or hybrid modeling approaches. In this study, we focused on comparing two classical, one machine learning, and one deep learning time series

---

## [Decision Letter · Decision Letter 1]

15 Dec 2025

Predicting the frequency of positive laboratory submissions for porcine reproductive and respiratory syndrome in Ontario, Canada, using autoregressive integrated moving average, exponential smoothing, random forest, and recurrent neural network

PONE-D-25-39776R1

Dear Dr.Tatiana Petukhova ,

We’re pleased to inform you that your manuscript has been judged scientifically suitable for publication and will be formally accepted for publication once it meets all outstanding technical requirements.

Kind regards,

Ahmad Salimi

Academic Editor

PLOS One

Additional Editor Comments (optional):

Reviewers' comments:

Reviewer's Responses to Questions

**Comments to the Author**

Reviewer #1: (No Response)

2. Is the manuscript technically sound, and do the data support the conclusions?

Reviewer #1: Yes

3. Has the statistical analysis been performed appropriately and rigorously?

Reviewer #1: Yes

4. Have the authors made all data underlying the findings in their manuscript fully available?

Reviewer #1: Yes

5. Is the manuscript presented in an intelligible fashion and written in standard English?

Reviewer #1: Yes

Reviewer #1: All comments from the previous review round have been addressed adequately. The manuscript is technically sound, the statistical analysis is appropriate, and the data support the conclusions. The language is clear and the manuscript is well-presented. I recommend acceptance.”

**Do you want your identity to be public for this peer review?** For information about this choice, including consent withdrawal, please see our Privacy Policy

Reviewer #1: No

---

## [Editor Report · Acceptance letter]

PONE-D-25-39776R1

PLOS One

Dear Dr. Petukhova,

I'm pleased to inform you that your manuscript has been deemed suitable for publication in PLOS One. Congratulations! Your manuscript is now being handed over to our production team.

Kind regards,

on behalf of

Dr. Ahmad Salimi

Academic Editor

PLOS One